# Effects of Gossypetin on Glucose Homeostasis in Diet-Induced Pre-Diabetic Rats

**DOI:** 10.3390/molecules29184410

**Published:** 2024-09-17

**Authors:** Karishma Naidoo, Andile Khathi

**Affiliations:** Department of Human Physiology, School of Laboratory Medicine and Medical Sciences, College of Health Sciences, University of KwaZulu-Natal, Durban 4000, South Africa; khathia@ukzn.ac.za

**Keywords:** gossypetin, flavonoids, diet-induced pre-diabetes, glucose homeostasis, natural compound

## Abstract

Natural flavonoids exert many potential health benefits, including anti-hyperglycaemic effects. However, the effects of gossypetin (GTIN) on glucose homeostasis in pre-diabetes have not yet been investigated. This study examined the effects of GTIN on key markers of glucose homeostasis in a diet-induced pre-diabetic rat model. Pre-diabetes was induced by allowing the animals to feed on a high-fat high-carbohydrate (HFHC) diet supplemented with 15% fructose water for 20 weeks. Following pre-diabetes induction, the pre-diabetic animals were sub-divided into five groups (*n* = 6), where they were either orally treated with GTIN (15 mg/kg) or metformin (MET) (500 mg/kg), both with and without dietary intervention, over a 12-week period. The results demonstrated that animals in the untreated pre-diabetic (PD) control group exhibited significantly higher fasting and postprandial blood glucose levels, as well as elevated plasma insulin concentrations and increased homeostatic model assessment for insulin resistance (HOMA2-IR) index, relative to the non-pre-diabetic (NPD) group. Similarly, increased caloric intake, body weight and plasma ghrelin levels were observed in the PD control group. Notably, these parameters were significantly reduced in the PD animals receiving GTIN treatment. Additionally, glycogen levels in the liver and skeletal muscle, which were disturbed in the PD control group, showed significant improvement in both GTIN-treated groups. These findings may suggest that GTIN administration, with or without dietary modifications, may offer therapeutic benefits in ameliorating glucose homeostasis disturbances associated with the PD state.

## 1. Introduction

Pre-diabetes represents a state of intermediate hyperglycaemia characterised by blood glucose levels that are elevated beyond normoglycaemia but do not meet the diagnostic criteria for type 2 diabetes mellitus (T2DM) [1]. This condition is associated with early insulin resistance and hyperinsulinaemia [2,3]. The prevalence of pre-diabetes has been linked to dietary patterns, particularly those high in fats and carbohydrates, commonly observed in “Westernized” diets [4]. As of 2017, the global prevalence of impaired glucose tolerance (IGT) was estimated at 7.3% among adults, affecting approximately 352.1 million individuals, with projections indicating an increase to 8.3% by 2045 [1,4]. Untreated pre-diabetes is recognised as a precursor to T2DM development and its associated complications [4,5]. The pathophysiology of pre-diabetes involves enhanced hepatic glucose storage and diminished glucose uptake by skeletal muscles [5]. Additionally, decreased cellular glucose uptake stimulates increased food intake, mediated by elevated ghrelin levels [6]. The progression from pre-diabetes to T2DM is significantly influenced by lifestyle factors, particularly dietary habits and adherence to prescribed interventions. Despite the proven efficacy of dietary modifications and pharmacological treatments in mitigating the risk of T2DM, adherence to these lifestyle changes remains a challenge [7]. Thus, there is a need for alternative treatments that remain effective even in the absence of dietary modifications [6]. Flavonoids, a diverse group of polyphenolic compounds found abundantly in plants, have demonstrated significant antioxidant properties, capable of scavenging various free radicals including hydroxyl radicals, superoxide anions and lipid peroxy radicals [8,9]. The antioxidant efficacy of flavonoids is influenced by their structural configuration and the number of hydroxyl groups present [10,11]. Quercetin, a well-known flavonoid, has been reported to improve glucose homeostasis in both pre-diabetic and diabetic states due to its antioxidant activity [12,13]. GTIN, a hexahydroxylated flavonoid extracted from the calyx of *Hibiscus sabdariffa*, possesses a chemical structure similar to that of quercetin [14,15]. It has been reported to exhibit more potent antioxidant effects compared to quercetin [14,15]. Previous research has highlighted that GTIN exhibits anti-inflammatory, nephroprotective, neuroprotective and hepatoprotective properties [14,15]. Despite this interesting finding, there is a lack of research into the potential of GTIN to modulate glucose homeostasis in PD conditions. Therefore, this study aims to evaluate the effects of GTIN on key markers associated with glucose homeostasis in a diet-induced PD rat model, both with and without dietary modifications.

## 2. Results

### 2.1. Oral Glucose Tolerance Test (OGTT)

The OGTT was measured at the end of week 12 of the treatment period (*n* = 6, per group). At 0 min (Figure 1a), the PD control group showed significantly (*p* < 0.05) higher fasting blood glucose (FBG) levels in comparison to the NPD group. However, PD animals receiving GTIN with either a normal diet (GTIN + ND) or a high-fat high-carbohydrate diet (GTIN + HFHC) showed significantly (*p* < 0.05) lower plasma glucose levels in comparison to the PD control group. Similarly, FBG levels were significantly (*p* < 0.05) lower in PD animals receiving MET with a normal diet (MET + ND) in comparison to the PD control group. At 120 min post-glucose load, the PD control group showed significantly (*p* < 0.05) higher plasma glucose levels in comparison to the NPD group. The GTIN + ND and GTIN + HFHC groups had significantly lower blood glucose levels by 37.72% and 27.09%, while the MET + ND and MET + HFHC groups showed reductions of 30.10% and 23.93% in comparison to the PD control group. Additionally, the PD control group exhibited a significantly higher area under the curve (AUC) at all intervals (Figure 1b) in comparison to the NPD group. Both GTIN-treated groups (31.16% and 19.32%) and MET-treated groups (25.30% and 3.45%) exhibited significantly lower AUC values at 60–120 min in comparison to the PD control group.

### 2.2. HOMA2-IR Index

The HOMA2-IR values were calculated at week 12 of the treatment period (*n* = 6, per group). The results (Table 1) showed that the FBG (*p* < 0.05) and insulin (*p* < 0.05) levels were significantly higher in the PD control group in comparison to the NPD group. The HOMA2-IR value for the NPD group was within the insulin-sensitive range (<1.0), while the PD control group had a significantly (*p* < 0.05) higher HOMA2-IR value compared to the NPD, which was in the range of early insulin resistance (>1.9). Both GTIN + ND and GTIN + HFHC groups showed significantly lower FBG (17.44% and 10.76%), insulin levels (57.88% and 53.05%) and HOMA2-IR index values (70.37% and 62.0%) in comparison to the PD control group. The MET + ND group also displayed significantly (*p* < 0.05) lower FBG (16.80%), insulin (57.20%) and HOMA2-IR values (60.07%) in comparison to the PD control group.

### 2.3. Glycated Haemoglobin

Glycated haemoglobin (HbA1c) concentrations were measured in all experimental groups at week 12 of the treatment period (*n* = 6, per group). The results (Figure 2) showed that the HbA1c levels were significantly (*p* < 0.05) higher in the PD control group in comparison to the NPD group. However, it was observed that the HbA1c levels were significantly (*p* < 0.05) lower in both the GTIN + ND and GTIN + HFHC groups by 54.24% and 50.22% in comparison to the PD control group. Similarly, both the MET + ND and MET + HFHC groups showed significantly lower HbA1c levels by 53.66% and 30.70% in comparison to the PD control group.

### 2.4. Caloric Intake

The caloric intake and percentage increase in caloric intake of the animals were determined every fourth week of the 12-week treatment period (*n* = 6, per group). The results (Table 2) showed that the PD control group had significantly (*p* < 0.05) higher caloric intake in comparison to the NPD group throughout the treatment period. At week 12, both the GTIN + ND and GTIN + HFHC groups showed significantly reduced caloric intake (17.26% and 11.86%) and percentage increase in caloric intake (19.18% and 6.76%) in comparison to the PD control group. The MET + ND group also exhibited significantly (*p* < 0.05) lower caloric intake (15.46%) and percentage increase (14.53%) in comparison to the PD control group.

### 2.5. Body Weight

The body weight and percentage increase body weight of the animals were determined every fourth week of the 12-week treatment period (*n* = 6, per group). The results (Table 3) showed that the body weight of the PD control group was significantly (*p* < 0.05) higher throughout the experimental period in comparison to the NPD group. At week 8, the GTIN + ND group demonstrated a significantly (*p* < 0.05) lower body weight and percentage increase (11.35% and 11.34%) in comparison to the PD control group. By week 12, reductions were 12.28% in body weight and 4.92% in percentage increase. The GTIN + HFHC group showed a significant (*p* < 0.05) reduction in percentage body weight increase by 4.59% at week 12 in comparison to the PD control group. The MET + ND group also exhibited a significantly (*p* < 0.05) lower percentage increase in body weight by 3.08% in comparison to the PD control group.

### 2.6. Plasma Ghrelin Concentrations

Plasma ghrelin concentrations were measured in all experimental groups at week 12 of the treatment period (*n* = 6, per group). The results (Figure 3) showed that the plasma ghrelin levels were significantly (*p* < 0.05) higher in the PD control group in comparison to the NPD group. Furthermore, both the GTIN + ND and GTIN + HFHC groups had significantly (*p* < 0.05) lower plasma ghrelin levels by 41.53% and 33.79% in comparison to the PD control group. Interestingly, there was no significant difference in the ghrelin concentrations between the NPD and GTIN + ND groups. Similarly, both the MET + ND and MET + HFHC groups had significantly (*p* < 0.05) lower plasma ghrelin by 40.29% and 13.51% in comparison to the PD control group.

### 2.7. Liver and Skeletal Muscle Glycogen Concentrations

Liver and skeletal muscle glycogen concentrations were measured in all experimental groups at week 12 of the treatment period (*n* = 6, per group). The results showed (Figure 4) that the glycogen concentrations in the PD control group were significantly (*p* < 0.05) higher in the liver, while significantly (*p* < 0.05) lower in the skeletal muscle in comparison to the NPD group. Both GTIN + ND and GTIN + HFHC groups showed significantly (*p* < 0.05) lower liver glycogen levels by 54.26% and 52.86% and significantly (*p* < 0.05) higher skeletal muscle glycogen levels by 60.40% and 47.66% in comparison to the PD control group. Similarly, the MET + ND and MET + HFHC groups exhibited significantly (*p* < 0.05) lower liver glycogen levels by 51.70% and 45.27% and higher skeletal muscle glycogen levels by 53.08% and 27.51% in comparison to the PD control group.

## 3. Discussion

Flavonoids are phenolic compounds found in fruits and vegetables as secondary metabolites [16]. These naturally occurring compounds possess anti-diabetic effects [10,17]. Quercetin, a well-known flavonoid, has been shown to significantly improve insulin sensitivity, reduce caloric intake, HbA1c levels and FBG in the PD state [10,18,19]. Thus, there is an increased demand to investigate quercetin and its derivatives, as it is hypothesised that similarly structured compounds may exhibit similar biological activity [20,21]. GTIN ameliorates many complications by targeting oxidative stress and inflammation [22,23]. However, the effects of this compound on glucose homeostasis in the PD state have not been explored. Hence, this study sought to investigate the effects of GTIN on selected glucose homeostasis-associated markers in PD rats, both with and without dietary modification.

Pre-diabetes is a state of intermediate hyperglycaemia associated with IGT, impaired FBG and early insulin resistance [1]. In normal glucose tolerant (NGT) individuals, blood glucose levels temporarily increase following glucose ingestion but return to fasting levels within two hours [24]. Insulin resistance impairs glucose disposal in skeletal muscle due to disrupted insulin signalling, glucose transport and glycogen synthesis, resulting in elevated blood glucose levels [25,26]. Consistent with previous studies, chronic consumption of a HFHC diet induced pre-diabetes, as evidenced by the impaired FBG, IGT and HOMA2-IR index in the PD control group in comparison to the NPD group [27,28,29]. However, both the GTIN + ND and GTIN + HFHC diet groups showed significantly lower HOMA2-IR index, fasting and postprandial glucose levels compared to the PD control group. This may suggest that GTIN has a beneficial effect on FBG levels and improves glucose tolerance even when combined with the HFHC diet, though its effects are more pronounced with a normal diet. The AUC was significantly lower in the GTIN + ND in comparison to the PD control group. This may suggest that GTIN improves glucose handling and reduces glucose exposure. Previous studies have shown that quercetin may stimulate glucose uptake, glucose transporter (GLUT)-4 translocation and insulin signalling pathways [30,31]. It is speculated that GTIN may improve glucose parameters by regulating key signalling pathways involved in glucose metabolism [32]. This improvement may have been due to increased glucose uptake in tissues and reduced FBG levels, leading to decreased beta-cell insulin secretion [32]. Additionally, the MET + HFHC and MET + ND groups exhibited similar therapeutic effects as the GTIN-treated groups, reinforcing the efficacy of MET in improving glucose tolerance. Both the MET- and GTIN-treated groups were effective in improving glucose tolerance in PD animals, with GTIN exhibiting more potent anti-hyperglycaemic effects.

HbA1c is a non-enzymatic glycation product of haemoglobin that persists in the body for 2–3 months [33]. Therefore, the HbA1c test is an effective means to screen for pre-diabetes and T2DM [33]. In the PD condition, plasma insulin does not effectively stimulate insulin-responsive tissues such as skeletal muscle [34,35]. This results in increased glycation of haemoglobin due to elevated plasma glucose levels [36]. In this study, at week 12 of the treatment period there were significantly higher HbA1c levels in the PD control group in comparison to the NPD group. The HFHC diet may have induced insulin resistance resulting in elevated HbA1c levels, consistent with previous studies [37,38]. However, both GTIN-treated groups showed significant reductions in HbA1c levels in comparison to the PD control group, with no change in comparison to the NPD group. GTIN was more effective at lowering HbA1c levels when combined with a normal diet compared to the HFHC diet, suggesting that a healthier diet enhances the efficacy of this compound. Previous studies have shown that flavonoids such as quercetin significantly improve HbA1c levels by increasing glucose uptake and improving insulin sensitivity [39,40]. Similarly, in this study, GTIN may have improved insulin-dependent glucose utilization, thereby reducing circulating plasma glucose and HbA1c levels. Additionally, MET has been shown to decrease HbA1c by decreasing hepatic glucose production and enhancing peripheral glucose uptake, as evidenced by the decreased HbA1c levels in the MET-treated groups [41,42]. However, GTIN appears slightly more effective than MET, particularly in the GTIN + ND group. The effect of GTIN was comparable to MET, indicating its potential as a promising alternative treatment in the management of diabetes.

Energy homeostasis is the state of equilibrium between energy intake and expenditure, which is impacted by caloric intake and energy storage [43]. High-fat diets (HFD) in rodents increase caloric intake, correlating with weight gain [44,45]. High-fructose diets stimulate appetite and body weight gain, impairing insulin sensitivity through elevated ghrelin secretion [44,45,46,47]. Ghrelin, an orexigenic hormone, regulates appetite and energy homeostasis [48]. Elevated ghrelin levels in insulin-resistant states stimulate increased food intake due to glucose deprivation in tissues [48]. In this study, at week 12 of the treatment period, the PD control group had significantly higher caloric intake, percentage body weight increase and plasma ghrelin levels in comparison to the NPD group. The results corroborated with previous findings which have associated elevated ghrelin with increased caloric intake and weight gain during pre-diabetes [49,50]. However, caloric intake, percentage body weight gain and ghrelin levels were significantly lower in both GTIN-treated groups at week 12 of the treatment period in comparison to the PD control group. These results may suggest that GTIN administration under both dietary regimens, significantly improves satiety and reduces body weight gain. Previous studies have shown that flavonoids inhibit body weight gain by reducing caloric intake and increasing satiety [51,52]. Similarly, GTIN may have reduced body weight gain by lowering caloric intake and decreasing ghrelin levels, potentially improving glucose uptake and glycogen synthesis. Additionally, the MET + ND group exhibited similar anti-obesity effects as the GTIN-treated groups. However, reductions in caloric intake and body weight were not significant in the MET + HFHC group. This may suggest that GTIN exhibits more potent effects on controlling obesity even in the absence of dietary intervention, as compared to MET administration. The results of this study may suggest that GTIN could serve as a potential anti-obesity drug.

Glycogen is the main form of glucose storage in the liver and skeletal muscles [53]. It is a readily available source of glucose for energy production [53]. Under normal physiological conditions, elevated plasma glucose stimulates insulin release, activating glycogen synthase and increasing glycogen synthesis in the liver and muscles [54,55]. In the PD state, skeletal muscle glycogen levels decrease due to impaired glucose uptake in insulin-resistant states, while excess glucose is diverted to the liver for increased glycogenesis [56,57]. In this study, skeletal muscle glycogen levels were significantly lower while liver glycogen levels were significantly higher in the PD control group in comparison to the NPD group. During the PD state, the excess plasma glucose in the presence of skeletal muscle insulin resistance may have been diverted to the liver for storage [58]. However, both GTIN-treated groups had significantly higher skeletal muscle glycogen and lower liver glycogen levels in comparison to the PD control group. Glycogen distribution is more favourable in the GTIN + ND group in comparison to the GTIN + HFHC group. Previous studies have shown that quercetin improves insulin signalling in skeletal muscle, thereby promoting increased glucose uptake and storage as glycogen [59,60]. Similarly, GTIN may have improved insulin signalling and glucose uptake in skeletal muscle, reducing glucose shunting to the liver. Additionally, the MET-treated groups showed similar effects on glycogen levels in the liver and muscle as the GTIN-treated groups. The two treatments were comparable in this regard.

The HFHC diet effectively induced pre-diabetes, as evidenced by impaired FBG, IGT, increased HbA1c, hyperinsulinaemia and early insulin resistance in comparison to the NPD group. However, the administration of GTIN, both in the presence and absence of dietary modification, significantly decreased FBG and HbA1c levels as well as improved glucose tolerance and insulin sensitivity in comparison to the PD control group. Furthermore, both GTIN-treated groups showed significantly lower caloric intake, body weight gain and plasma ghrelin levels in comparison to the PD control group. GTIN improved disturbed glycogen levels in the liver and skeletal muscle. Additionally, the effects of GTIN were comparable to the conventional anti-diabetic drug MET. This may suggest that GTIN, with or without dietary modification, ameliorates disturb glucose homeostasis in the PD state. Further studies are needed to ascertain the mechanism of the anti-hyperglycaemic actions of GTIN.

## 4. Materials and Methods

### 4.1. Chemicals and Drugs

All chemicals and drugs used were of analytical grade and obtained from reputable suppliers (Merck chemicals (PTY) LTD, Wadeville, Gauteng, South Africa).

### 4.2. Animals and Housing

This study involved the use of 36 male Sprague-Dawley rats (150–180 g) which were bred and housed in the Biomedical Research Unit (BRU) at the University of KwaZulu-Natal (UKZN), Westville campus. The animals were maintained under controlled laboratory conditions: a constant temperature of 22 ± 2 °C, carbon dioxide (CO_2_) levels below 5000 ppm, relative humidity of 55 ± 5%, and a 12-h light/dark cycle with lights on at 07:00. Noise levels were kept below 65 decibels, and the rats had ad libitum access to food and water. All protocols and animal care procedures adhered to the guidelines of the Animal Research Ethics Committee of the University of KwaZulu-Natal (ETHICS: AREC/0000495/2022). Prior to the study, the rats were acclimatized to their environment for one week with standard rat chow and tap water before exposure to the HFHC diet (AVI Products (Pty) Ltd., Waterfall, South Africa) [50]. 

### 4.3. Induction of Pre-Diabetes

The rats were randomly assigned to two diet groups: Group 1 (*n* = 6) and Group 2 (*n* = 30) (Figure 5). Pre-diabetes was induced in Group 2 according to the protocol described by Luvuno et al. [50]. Group 1 rats were maintained on a standard rat diet with tap water, while Group 2 rats were given a HFHC diet supplemented with 15% fructose-enriched water. After 20 weeks, pre-diabetes was assessed using the American Diabetes Association (ADA) criteria [61]. Rats with FBG levels between 5.6 to 6.9 mmol/L, an OGTT 2-h glucose concentration between 7.8 to 11.0 mmol/L, and HbA1c levels between 5.7 to 6.4% were classified as pre-diabetic. Rats fed the standard diet were also evaluated at week 20 to confirm normoglycemia and were classified as the non-pre-diabetic group. This period marked the beginning of the treatment phase (Week 0).

### 4.4. Experimental Design and Treatment

Following pre-diabetes induction, the rats were categorized into two primary groups: non-pre-diabetic (*n* = 6) and pre-diabetic (*n* = 30). The treatment phase commenced the day after pre-diabetes diagnosis and continued for 12 weeks. The pre-diabetic rats (*n* = 30) were subdivided into five experimental groups (Groups 2 to 6), each comprising six rats. These groups were as follows: Group 2 (pre-diabetic control) continued on the HFHC diet and received as the vehicle, 3 mL/kg of diluted dimethyl sulfoxide (DMSO) (2 mL DMSO: 19 mL normal saline, p.o.); Group 3 (GTIN + ND) received GTIN while on the normal diet; Group 4 (GTIN + HFHC) received GTIN while continuing on the HFHC diet; Group 5 (MET + ND) received MET (MET) while on the normal diet; and Group 6 (MET + HFHC) received MET while on the HFHC diet. GTIN and MET were administered orally at dosages of 15 mg/kg and 500 mg/kg, respectively, every third day at 09:00 am. These dosages were selected based on previous studies [13,49,62]. Parameters such as body weight, caloric intake, and FBG levels were monitored at week 0, 4, 8, and 12. The OGTT was conducted at week 12 following the protocol established in prior studies [49,50].

### 4.5. Blood Collection and Tissue Harvesting

After 12 weeks of treatment, all animals were anaesthetised with Isoflurane (100 mg/kg) (Safeline Pharmaceuticals (Pty) Ltd., Roodeport, South Africa) for 3 min using a gas anaesthetic chamber (BRU, UKZN, Durban, SA). Following the onset of anaesthesia, blood samples were collected via cardiac puncture and transferred into pre-cooled heparinized tubes. The samples were then centrifuged (Eppendorf centrifuge 5403, Hamburg, Germany) at 4 °C and 503 g for 15 min to separate the plasma, which was subsequently stored at −80 °C in a Bio Ultra freezer (Snijers Scientific, Tilburg, The Netherlands) until ready for biochemical analysis. The liver and skeletal muscle tissues were excised after the animals were sacrificed, snap-frozen in liquid nitrogen, and stored at −80 °C in a BioUltra freezer until ready for glycogen assay.

### 4.6. Biochemical Analysis

Plasma insulin, HbA1c and ghrelin levels were measured using ELISA kits (Elabscience Biotechnology Co., Ltd., Houston, TX, USA) as per the manufacturer’s instructions. The HOMA2-IR index was calculated from the insulin and FBG concentrations https://www.dtu.ox.ac.uk/homacalculator/ (accessed on 8 July 2024).

### 4.7. Glycogen Assay

Glycogen levels in liver and skeletal muscle tissues were assessed following established protocols [62,63,64]. Tissue samples (50 mg) were homogenized with 2 mL of 30% potassium hydroxide (KOH) and heated at 100 °C for 30 min. The reaction was terminated by the addition of 0.194 mL of 10% of sodium tetraoxosulphate VI (Na_2_SO_4_). After cooling, the glycogen precipitate was collected, washed with ethanol (95%), and resolubilized in 1 mL of H_2_O. The glycogen content was then quantified by mixing 200 µL of the resolubilized sample with 4 mL of anthrone reagent (0.5 g anthrone in 250 mL of 95% sulfuric acid) and boiling for 10 min. Absorbance was measured at 620 nm using a Spectrostar Nano spectrophotometer (BMG Labtech, Ortenburg, Germany).

### 4.8. Statistical Analysis

Data are expressed as mean ± SEM. Statistical analyses were performed using one-way analysis of variance (ANOVA) followed by Tukey–Kramer post hoc tests. AUC calculations were conducted using GraphPad Prism 5 software. Statistical significance was determined at *p* < 0.05.

## 5. Conclusions

In conclusion, this study establishes GTIN as a promising agent for improving glucose homeostasis in PD rats. GTIN significantly reduced FBG, enhanced glucose tolerance and lowered HbA1c levels. It also improved metabolic parameters by reducing caloric intake, body weight gain and plasma ghrelin levels, suggesting its benefits in appetite control and weight management. Additionally, GTIN positively influenced glycogen distribution, enhancing glucose uptake in muscles and reducing liver glycogen. These effects were observed with and without dietary changes, with more pronounced benefits on a normal diet. The efficacy of GTIN in glucose management was comparable to that of MET, indicating its potential as an alternative therapeutic agent in diabetes management.

## Figures and Tables

**Figure 1 molecules-29-04410-f001:**
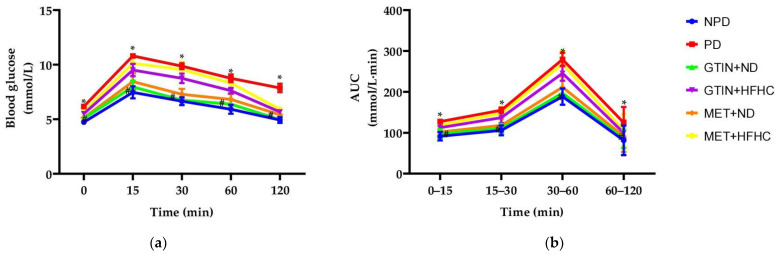
The (**a**) blood glucose levels during 2-h OGTT and (**b**) AUC values for the time intervals indicated following glucose load in the respective groups at week 12 of the treatment period (*n* = 6, per group). Values are expressed as mean ± SEM. * = *p* < 0.05 denotes comparison to NPD, # = *p* < 0.05 denotes comparison to PD.

**Figure 2 molecules-29-04410-f002:**
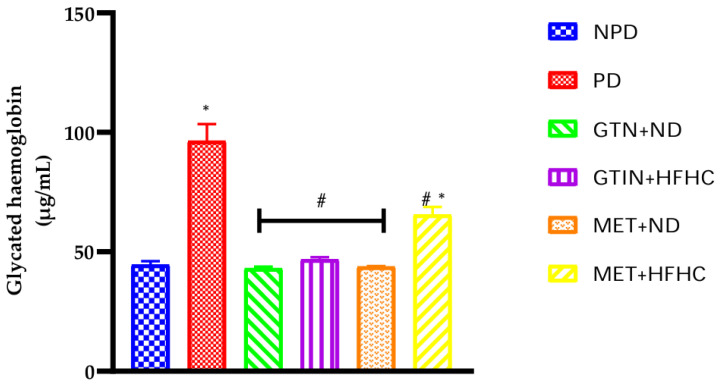
The effects of GTIN on the HbA1c levels in rats with or without dietary modification at week 12 of the treatment period (*n* = 6, per group). Values are presented as mean ± SEM.* = *p* < 0.05 denotes comparison with NPD; # = *p* < 0.05 denotes comparison with PD.

**Figure 3 molecules-29-04410-f003:**
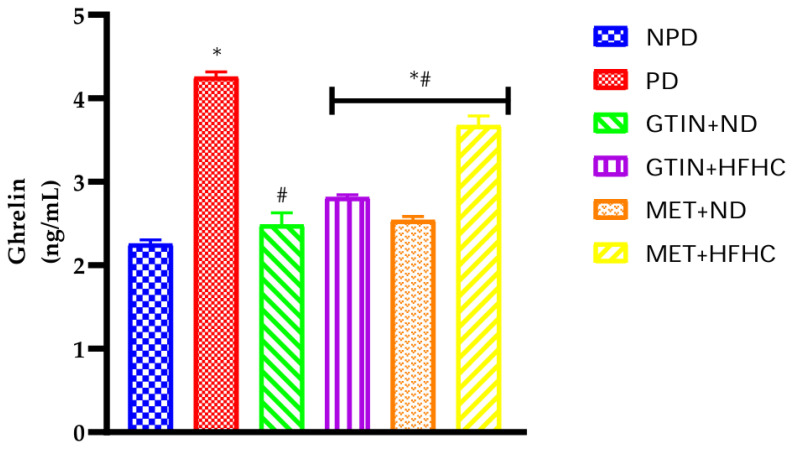
The effects of GTIN on the plasma ghrelin concentration in rats with or without dietary modification at week 12 of the treatment period (*n* = 6, per group). Values are presented as mean ± SEM. * = *p* < 0.05 denotes comparison with NPD; # = *p* < 0.05 denotes comparison with PD.

**Figure 4 molecules-29-04410-f004:**
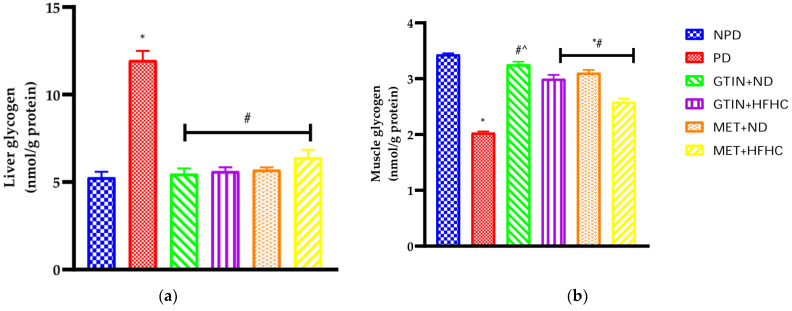
The effects of GTIN on (**a**) liver and (**b**) skeletal muscle glycogen concentrations in rats with or without dietary modification at week 12 of the treatment period (*n* = 6, per group). Values are presented as mean ± SEM in each group (*n* = 6). * = *p* < 0.05 denotes comparison with NPD; # = *p* < 0.05 denotes comparison with PD; ^ = *p* < 0.05 denotes comparison with MET + HFHC.

**Figure 5 molecules-29-04410-f005:**
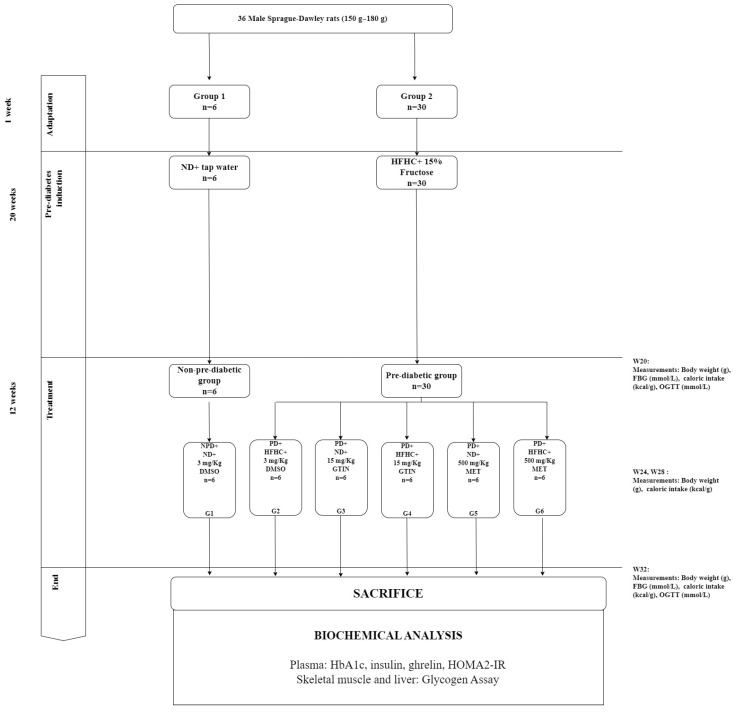
Layout of experimental design and grouping.

**Table 1 molecules-29-04410-t001:** The effects of GTIN on FBG, plasma insulin and HOMA2-IR index in rats with or without dietary modification at week 12 of the treatment period (*n* = 6, per group). Values are presented as mean ± SEM.

Experimental Group	Plasma Fasting Glucose (mmol/L)	Plasma Insulin (pmol/L)	HOMA2-IR Index
NPD	4.73 ± 0.12	28.03 ± 0.79	0.85 ± 0.038
PD	6.17 ± 0.08 *	68.36 ± 4.09 *	2.74 ± 0.19 *
GTIN + ND	5.08 ± 0.14 # ^	28.22 ± 0.83 #	0.92 ± 0.036 #
GTIN + HFHC	5.50 ± 0.073 #	31.57 ± 0.43 #	1.11 ± 0.022 #
MET + ND	5.13 ± 0.12 #	28.70 ± 0.66 #	0.94 ± 0.034 #
MET + HFHC	5.70 ± 0.073 *	34.51 ± 1.30 #	1.26 ± 0.063 *

* = *p* < 0.05 denotes comparison with NPD; # = *p* < 0.05 denotes comparison with PD; ^ = *p* < 0.05 denotes comparison with MET + HFHC.

**Table 2 molecules-29-04410-t002:** The effects of GTIN on caloric intake and percentage increase in caloric intake from week 0 to week 12 of the treatment phase in rats with or without dietary modification (*n* = 6, per group). Values are presented as mean ± SEM and in percentage increase (↑ = increase and ↓ = decrease).

Caloric Intake (Kcal/g)				
Experimental Group	Week 0	Week 4	Week 8	Week 12
NPD	139.92 ± 2.25 (100%)	142.90 ± 2.44 (↑ 2.15%)	144.34 ± 2.23 (↑ 3.14%)	144.16 ± 2.12 (↑ 3.02%)
PD	162.14 ± 2.64 * (100%)	171.16 ± 2.39 * (↑ 5.28%) *	169.43 ± 2.74 * (↑ 4.30%)	170.93 ± 2.99 * (↑ 5.13%)
GTIN + ND	168.13 ± 2.84 * (100%)	160.47 ± 2.79 * (↓ 4.77%) *#^	150.87 ± 1.98 (↓ 11.42%) *#^	141.10 ± 2.62 #^ (↓ 19.18%) *#^
GTIN + HFHC	163.31 ± 4.37 * (100%)	165.08 ± 4.51 * (↑ 2.98%) #	166.06 ± 4.78 * (↑ 3.09%)	150.65 ± 3.64 # (↓ 6.76%) *#
MET + ND	165.12 ± 1.66 * (100%)	162.09 ± 1.67 * (↓ 1.87%) *#^	159.02 ± 1.89 * (↓ 2.12%) *#^	144.18 ± 1.56 #^ (↓ 14.53%) *#^
MET + HFHC	162.79 ± 2.65 * (100%)	168.86 ± 2.92 * (↑ 3.58%)	168.45 ± 2.04 * (↑ 3.39%)	158.17 ± 2.55 * (↓ 2.92%) *#

* = *p* < 0.05 denotes comparison with NPD, # = *p* < 0.05 denotes comparison with PD, ^ = *p* < 0.05 denotes comparison with MET + HFHC.

**Table 3 molecules-29-04410-t003:** The effects of GTIN on body weight and percentage increase in body weight from week 0 to week 12 of the treatment phase in rats with or without dietary modification (*n* = 6, per group). Values are presented as mean ± SEM and in percentage increase (↑ = increase and ↓ = decrease).

Body Weight (g)				
Experimental Group	Week 0	Week 4	Week 8	Week 12
NPD	528.2 ± 18.21 (100%)	540.7 ± 18.53 (↑ 2.30%)	554.0 ± 17.87 (↑ 4.65%)	554.8 ± 17.51 (↑ 4.77%)
PD	650.2 ± 9.46 * (100%)	673.7 ± 11.46 * (↑ 3.47%)	692.7 ± 10.17 * (↑ 6.13%)	708.0 ± 15.71 * (↑ 8.09%)
GTIN + ND	665.3 ± 16.86 * (100%)	634.0 ± 15.08 * (↓ 8.63%) *#^	612.7 ± 17.04 # (↓ 11.34%) *#^	597.7 ± 15.74 #^ (↓ 4.92%) ^#*
GTIN + HFHC	633.0 ± 13.09 (100%)	637.0 ± 12.96 #^ (↑ 0.67%)	651.5 ± 12.21 # (↑ 2.83%)	663.0 ± 11.88 (↑ 4.59%) #^
MET + ND	638.5 ± 18.71 * (100%)	612.7 ± 18.78 #* (↓ 4.24%)	618.8 ± 18.19 #^* (↓ 3.18%)	619.5 ± 18.73 (↓ 3.08%) *#
MET + HFHC	651.3 ± 23.82 * (100%)	671.3 ± 24.47 (↑ 2.98%)	690.7 ± 22.40 * (↑ 5.75%)	700.5 ± 23.49 * (↑ 7.06%) *

* = *p* < 0.05 denotes comparison with NPD; # = *p* < 0.05 denotes comparison with PD; ^ = *p* < 0.05 denotes comparison with MET + HFHC.

## Data Availability

The data presented in this study are available on request from the corresponding author.

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
