# Peer review of "Effects of Gossypetin on Glucose Homeostasis in Diet-Induced Pre-Diabetic Rats"

_molecules, 2024, doi:10.3390/molecules29184410_

Round 1
Reviewer 1 Report
Comments and Suggestions for Authors
Dear Authors,
The article “Investigating the therapeutic effects of gossypetin on glucose homeostasis in diet-induced pre-diabetic male Sprague Dawley rats” I was given to review is very interesting study of the potential benefitial effect of natural-derived compound on diet-induced ore-diabetic rodent model. This metabolic state is loaded with extremely serious social significance. In this regard, research in this area is useful to society as a whole, especially when substances are as widely distributed and readily available as the one under study.
However, I believe that the article could be published after major revision.
Here are my recommendations, as well as some inaccuracies and omissions, as follows.
1) I recommend that authors review the entire text, including the abstract, for possible repetitions. The description of the results attracted particular attention. Also, it would contribute to the quality of the publication if the verb tense used throughout the text is checked – there are gaps in some places. Correct everywhere the expression "by comparison" you have used with the correct "in comparison" and for other similar omissions.
2) Also, many abbreviations are not explained when they are first mentioned in the text, but only in the "Materials and methods" section
3) I have to give the authors a serious note about "measurement of AUC" as they describe it in their manuscript (line 67).
AUC is defined as the area under the concentration/time curve for a particular substance. I see two main problems here. First, the authors tracked a change in some concentration (presumably blood glucose) over time up to 120 minutes, but this was not a determination of AUC. The area under the concentration / time curve is calculated mathematically by a certain formula or with specialized software, and is used for pharmacokinetic studies that help determine the bioavailability of a given drug or substance depending on its route of administration. Second, looking at Figure 1, I noticed that the graphs are most likely swapped. In my opinion, A) represents the change in blood glucose levels over time in the animals of the different groups, and B) represents the state at the second hour of the OGTT.
4) It would be good if the authors clearly state which index they use in their calculations - HOMA-IR or HOMA2-IR, because they use both in the text (in the title of section 2.2. HOMA2-IR is indicated, and in the text below - HOMA-IR) [Song YS, Hwang YC, Ahn HY, Park CY. Comparison of the Usefulness of the Updated Homeostasis Model Assessment (HOMA2) with the Original HOMA1 in the Prediction of Type 2 Diabetes Mellitus in Koreans. Diabetes Metab J. 2016 Aug;40( 4):318-25. doi: 10.4093/dmj.2016.40.4.318. Epub 2016 May 27].
5) I believe it would be helpful if the description of the results is further expanded. Currently, the results are presented in a uniform manner, indicating only the direction of change of the studied indicator, and whether there is statistical significance. Completing, for example, percentages will help to compare the effects observed across groups better.
6) In my opinion, the information on gossypetin presented in the Discussion section (lines 226-230) would have contributed to the introduction, clarifying the interest in this substance.
7) Comparing the effects of gossypetin with those of the reference metformin is quite straightforward. Likewise, there is no comparison and corresponding conclusion regarding the observed effects of gossypetin in the HFHC and ND groups.
8) Regarding the description of materials and methods, I have the following notes:
- Section 4.2 does not state the total number of rats used in the study.
- I recommend that the authors present an outline of the experimental design to make it clearer to the reader the period of induction of the pre-diabetic state, the time of treatment, the diet during this time, and the time points for testing.
- On line 414, the authors have indicated the anesthetic used. Please, in accordance with the rules, use the international non-proprietary name (INN) of the pharmacological agent. Of course, the trade name, along with the batch and expiration date, can be specified for clarification.
- Section 4.5 does not state when the tissues were taken. I assume this was done after killing the rats, but after the blood was drawn under anesthesia, what happened next is not described.
- I believe there was a technical error on line 399. It should probably be "Group 2 to Group 6" instead of "Group 3 to Group 6". Otherwise, there is a contradiction with the description of the groups and the stated number of animals.
- In lines 419 and 421 degrees Celsius are not written correctly as on lines 434 the chemical formula needs to be corrected to Na2SO4, and 438 the chemical formula of water needs to be corrected to H2O.
9) The conclusion part should be revised and expanded to suit the serious and thorough research that has been done.
Finally, I leave it to the decision of the authors whether to shorten the title by removing some unnecessary clarifications to make it clearer.
Review date
Aug 7th 2024
Comments on the Quality of English Language
I recommend that authors review the entire text, including the abstract, for possible repetitions. The description of the results attracted particular attention. Also, it would contribute to the quality of the publication if the verb tense used throughout the text is checked – there are gaps in some places. Correct everywhere the expression "by comparison" you have used with the correct "in comparison" and for other similar omissions.
Reviewer 2 Report
Comments and Suggestions for Authors The authors presented interesting results of the effects of gossypetin on some parameters of pre-diabetes induced in male rats. However, the paper has huge similarity rates (the amount of wording duplication in the manuscript) and before any detailed review whole manuscript should be re-written. This is especially applicable for section Introduction and Methods where lot of whole sentences are copy/paste from another papers. Also, it looks like in hurry section Methods were moved below the Discussion and text was not updated to follow new order of Sections. Other comments: The second part of abstract should be re-written (starting from results). I do not understand what meaning comparasion to PD (Ln21) where all groups were PD. The same is applicable for all sentences in result section of Abstract. Also, abbreviation NPD is not introduced.
Ln60 and further on- if you introduced abbreviation for gossypetin please continue to use it
Ln66 it is better to use abbreviation OGTT instead of OGT
In Section 2 please introduce the abbreviation for PD, FBG etc. It looks like section Material and Methods are moved at the end of manuscript (per journal requirements), but the section Results is not update in order to make possible to follow up written text.
How old were used rats at the beginning of experiments?
Ln69 what is differences between abbreviation ND and NPD?
Why you mentioned AUC in Result section, while you do not mentioned in Methods section? Also, HOMA2-IR index should be explain in Methods section not in Results. And did you calculate HOMA2-IR or HOMA-IR?
Ln98 what meaning sign "=" before p value? The same is applicable for other figures as well
Ln223-224 why again you introduced same abbreviation FBG and ND?
Ln236 again and again PD already introduced so many times..... again same for IGT (Ln237).... and again in whole Discussion the abbreviation for gossypetin was not used....
Ln410 FBG already introduced, the same is applicable for OGTT at Ln411
Ln416 UKZN was not introduced while BRU was introduced but again repeat at Ln466-467
Ln423-424 already mentioned manufacturer of Bio Ultra freezer in sentence before and in Ln 422 it should be written the Netherlands instead of Holland
Round 2
Reviewer 1 Report
Comments and Suggestions for Authors
Dear Authors,
Thank for respecting my recommendations and indicated omissions. I am satisfied with the corrections and additions made.
I only disagree with the presentation of Fig. 1 B, as I already indicated in the first review. Ms Karishma Naidoo has justified the chosen way of visualization as made by GraphPad Prism 5 software. However, as a pharmacologist, I do not agree that this is the best way because Area under the concentration/time curve (AUC) represents the change in plasma concentration over a specific time period. Since this is obviously a time-varying quantity, it is not correct to represent it as a single value. I have included here a picture of what an AUC study should look like for the convenience of authors and recommend that they check the settings of the statistical program they used.

https://images.app.goo.gl/vNZJwQSEdqG7fhS46
In addition, I draw attention to the fact that the same figure (1 B) lacks a title on the “x” axis.
Reviewer 2 Report
Comments and Suggestions for Authors
Dear authors, I am satisfied with manuscript improvement and provided feedback.
Author Response
Dear Reviewer 2
Thank you for your feedback and for taking the time to review our manuscript. We are pleased to hear that you are satisfied with the improvements and feedback provided.
Best regards,
Ms Karishma Naidoo